# Active tuberculosis in household contacts of bacteriologically confirmed pulmonary tuberculosis patients: A multicenter study finding the 'Missed One' in Central Ethiopia

**Getachew Seid**[1,2]*, **Ayinalem Alemu**[1,2], **Getu Diriba**[1], **Michael Hailu**[1], **Amanuel Wondimu**[1], **Mengistu Tadesse**[1], **Gemechu Tadesse**[1], **Solomon H Mariam**[2], **Balako Gumi**[2]

**1** Ethiopian Public Health Institute, Addis Ababa, Ethiopia, **2** Aklilu Lemma Institute of Pathobiology, Addis Ababa University, Addis Ababa, Ethiopia

\* gech1365@gmail.com

## Abstract

### Background

There was a 'missing millions' gap between the incidence of tuberculosis (TB) cases and the notified cases. In many TB high-burden countries, only about 25% of household contacts (HHCs) completing household TB evaluation and 20–89% of eligible contacts did not adhere to TB screening. The study was conducted to assess the yield of door-to-door TB household contact investigation among household contact of bacteriologically confirmed pulmonary TB cases in central Ethiopia.

### Methods

This cross-sectional study was carried out in selected health facilities of central Ethiopia from January 1, 2023 to December 3, 2023.All sequential voluntary bacteriologically confirmed TB patients and their HHCs without discrimination by age were included in the study. Xpert Ultra assay and TB culture were used to investigate active TB from sputum sample. Spearman's correlation analysis was used to determine the correlation between the index case cycle threshold value and the corresponding HHCs. Multivariable logistic regression analysis was done to investigate the associated risk factors for active TB in HHCs.

### Results

Among 967 HHCs claimed by 303 index cases (259 drug susceptible TB (DS-TB) and 44 multi-drug resistance TB (MDR/RR-TB)), 902(93.07%) HHCs had received baseline symptom-based TB evaluation. Presumptive TB was identified in 20.17% (182) of the evaluated HHCs and 13(1.44%) were diagnosed with active TB. Eleven HHCs (7.24%; 95% CI: 3.85–12.9) from DS-TB index case contacts and 2 (6.67%; 95% CI: 1.16–23.51) from MDR/RR-TB indexes HHCs were found to be MTB detected Rifampicin resistance not detected cases. The Xpert ultra assay results revealed an 84.62% (95% CI: 57.77–95.68)

**Data availability statement:** All relevant data are within the manuscript and its Supporting Information files.

**Funding:** The author(s) received no specific funding for this work.

**Competing interests:** The authors have declared that no competing interests exist.

**Abbreviations:** AOR, Adjusted Odds Ratio; CI, Confidence Interval; HEW, Health Extension Workers; HHC, Household contacts; PTB, Pulmonary Tuberculosis; TB, Tuberculosis; WHO, World Health Organization.

Rifampicin drug resistance concordance between the index case and the corresponding HHC. Active TB was significantly associated with night sweating and sharing a bed with the index patient, P-value < 0.05.

## Conclusion

Home-to-home TB contact screening have high active TB yield and implementable in both rural and urban areas of the nation only by mentoring and motivating the health extension workers. Proximity to bacteriologically confirmed TB patient for long time exposes household contacts for active TB. Scheduling convenient times and last-mile service delivery to contacts is very important to address the missed active TB cases in the community.

## Introduction

Worldwide in 2023, 10.8 million people developed tuberculosis (TB), the disease is estimated to have killed 1.25 million people. It proceeds to be the second most frequent cause of mortality from a single infectious pathogen. According to 2025 WHO end TB milestone, there was a 'missing millions' gap between the incidence of TB cases and the notified TB cases. The bulk of missing persons with TB live in low-income countries, and many of them continue to have symptoms without seeking medical attention [1].

Active (TB) refers to a disease that occurs in a person infected with *Mycobacterium tuberculosis* (Mtb) and it occurs when the immune system cannot defend against an infection [2]. Once an individual is newly diagnosed with TB, the most effective procedure for managing household contacts (HHCs) includes screening for TB, treating any individual who has TB disease, and giving TB preventive treatment (TPT) to those with TB infection but not diseased based on eligibility criteria. This improves the early identification and management of contacts with disease. This recommended strategy is active TB contact investigation [3].

The outcome of TB by active case finding is reliant on the screening algorithm, the characteristics of the contacts being evaluated, and most importantly, the linkage between effective diagnostic method and treatment facilities [4]. The routine strategy to contact investigation is that the index patient is requested to declare the number of HHCs and bring all of them to the health facility for symptom evaluation and TPT initiation, if eligible. This passive method is hampered by several factors, including extended waiting time, difficulties with scheduling or finances, and reluctance by parents or health professionals towards screening and commencing TPT in a healthy child [5].

The main predictors of tuberculosis but not limited to are poverty, undernourishment, HIV infection, smoking, and diabetes [1]. The risk of TB among contacts is determined by the characteristics of the index cases and contacts, or the environment setting where the exposure happened. Bacteriological confirmation of TB and cavitary spots in the index patient could facilitate the release of large quantities of bacteria and increase the likelihood of transmission [6]. Many publications have shown an association between the bacillary load in sputum and Mtb transmission [7,8,9]. Close contacts of MDR-TB patients are more likely to contract drug resistance TB (DR-TB). Meanwhile, contradicting evidence has been reported from many studies regarding the risk of TB in close contacts of drug-susceptible and MDR-TB patients [8,9,10].

In many TB high-burden countries contact investigation access and uptake was poor, with only about 25% of household contacts completing household TB evaluation and 20–89% of eligible contacts did not adhere to TB screening [11,12]. The key bottlenecks to the uptake and adherence of contact investigation include a lack of TB-specific awareness; the stigma

associated with the disease; catastrophic costs; and dissatisfaction with the quality of health facility general services [13,14].

The most effective way to curve the incidence rate is to reach the most at-risk group. This study closes this gap by bringing TB contact investigation service closer to people in need thereby subsidizing catastrophic costs, delayed diagnosis, and treatment delay. The present study was conducted to study the yield of door-to-door TB household contact investigation among (HHC) of bacteriologically confirmed pulmonary TB cases in central Ethiopia.

## Methods

### Study design and setting

This cross-sectional study was conducted in central Ethiopia from January 1, 2023 to December 3, 2023. It includes a 200-kilometer radius from the capital Addis Ababa. It covers three zones of the Oromia region, the Addis Ababa city administration, one zone from Amhara region, and one zone from the central Ethiopia region. This setting is among the densely populated areas in Ethiopia, with an average of 398.4 persons per km2 and an average family size of 3.1 [15]. In Ethiopia TB care and management is decentralized to primary health care facility which meets prerequisites to give the service. In the study area, 42 public health hospitals and 372 health centers provide TB diagnostic and treatment services [16].

### Tuberculosis case identification and management in Ethiopia

In Ethiopia, hospitals or health centers are the facilities where the first TB diagnosis and treatment initiation take place. After that, patients are referred back to the health posts that are nearest to their place of residence for the remainder of their care. Health extension workers (HEWs) ensure adherence to treatment and follow-up through daily observation. HEW routinely visits households under their catchment area. During their visit, they will identify and refer presumptive TB cases, trace treatment interrupters and lost to follow-up, give health education, and perform household contact screening [17].

### Sample size calculation

The sample size was calculated using a single-population proportion formula with a 5% level of significance and a 1% margin of error.

According to earlier systematic reviews and meta-analyses, at baseline, 2.3% of household contacts screened had TB [18]. Hence, using these figures, the sample size was calculated to be 856, and including a 10% non-response rate the final sample size was 942.

Since it is not feasible to directly sample HHCs of TB patients, index TB cases were selected and all eligible HHCs were enrolled in the study. According to the national TB program, there were 3.1 HHCs per index TB patient. Based on these data we recruited 303 bacteriologically confirmed pulmonary TB patients without discrimination by drug resistance profile.

### Sampling procedure

Initially, a list of all TB diagnostic and treatment health facilities in the study area was obtained from the Ministry of Health. Based on the previous year TB caseload sites were grouped into high, medium, and low. Using a simple random method three sites were selected from each category. In central Ethiopia, there are seven drug-resistant tuberculosis (DR-TB) treatment initiating centers (TICs). We selected five of them randomly. The total sample size was proportionally distributed to each site. All consecutive voluntary bacteriologically confirmed PTB index patients and their HHCs were included in the study.

## Operational definition of variables

An active pulmonary TB case is a TB case confirmed by either smear microscopy, Gene Xpert MTB/RIF Ultra, or a TB culture test. Bacteriologically confirmed pulmonary tuberculosis: case referred to a pulmonary TB patient with biological specimen positive by Acid-Fast Bacilli (AFB) smear microscopy, Xpert MTB/RIF assay or TB culture, indifference with drug susceptibility profile. A household contact was defined as a person who shared the same enclosed living space as the index case for one or more nights or for frequent or extended daytime periods during the three months before the start of TB treatment. An index case was the first bacteriologically confirmed TB case in a household at any age that lived with at least one other person.

## Study participants

Every consented consecutive bacteriologically confirmed pulmonary TB patient (index cases) who had visited TB clinics at any age, had given a traceable residence location, and had one or more household contacts was eligible for the study. The study included index cases without discriminating by drug resistance profile. The study did not include prison index cases due to restricted entry.

As long as a HHC had lived with the infectious TB index case for at least three months, had not received TB treatment at or before the time of the home visit, had given their consent to the study, and was available for interviews during the home visits; they were included in the study regardless of age.

## Contact investigation procedure

This study executed a nationally recommended active TB contact investigation strategy for the evaluation and management of HHCs at the dwelling. Once an individual is diagnosed with bacteriologically confirmed pulmonary TB (index case), the health personnel in the TB clinic request the index case to list all household contacts that live with him in the same dwelling. The guardians or parents of children index case, under 18 years, were asked for their consent. If two or more index cases were not diagnosed on the same day; one index case means one household. The head of the household was asked whether s/he is voluntary to host HEW in his/her dwelling for contact TB symptom evaluation. The HEW visits the household in the morning section of the scheduled date.

On the booked date the HEWs asked verbal permission to give health education related to TB and conduct TB symptom screening. When HHCs missed the first visit second round appointment was made to address them. If they missed the second time they were counted as missing the evaluation. Household contacts having contact with the index case were examined for active TB through symptom screening. A presumptive TB case was a contact with cough of two weeks or more or having any two of the following symptoms: fever of two weeks or more, night sweats, and unexplained weight loss of more than 1.5 kg in a month. Index cases were requested to give sputum samples before they started TB treatment. Tuberculosis symptomatic HHCs gave early morning sputum for bacteriological confirmation of active tuberculosis. Those who were unable to produce appropriate samples were advised to try during the second round of home visits. Tuberculosis-asymptomatic HHCs were advised to come to health posts or health facilities whenever they experience TB symptoms (S1 Fig).

## Microbiological evaluation for active TB cases

The collected samples were transported to the Ethiopian Public Health Institute using cold chain courage (2–8ºc) through the postal system within the collection date. The samples were examined for TB using Xpert Ultra assay, smear microscopy, and Mtb culture. The result

was communicated to the health facility clinician who sent the sample immediately after the release of the result. The principal investigator and the clinician followed the initiation of TB treatment for active TB-positive household contacts. Under 15 years of children (contacts of drug-susceptible index case) who ruled out TB were linked to TPT.

All specimens were processed using the NALC-NaOH (N-acetyl L-cystine sodium hydroxide sodium citrate) digestion decontamination technique described in the Global Laboratory Initiative (GLI) manual (19). Following processing, 0.5 ml was added to liquid media Mycobacteria Growth indicator tube (MGIT) and 0.1 ml inoculated to LJ(Lowenstein Jenson) media, AFB(Acid fast bacilli) smear was prepared and stained with the Zehil Nelsson (ZN) Staining method. Tuberculosis culture is highly sensitive with lower detection limit of 10–100 CFU/ml. Xpert Ultra assay was done within 24 hours after the arrival of the samples at the reference laboratory. Xpert Ultra assay has a sensitivity of 90% (95%CI: 84–94) and specificity 96%(95%CI:93–98).The result interpretation and test procedure were done based on the test user manual guide [19,20].

## Data collection and quality assurance

A data collection tool was developed to collect socio-demographic characteristics, clinical findings, risk factors for active TB, and laboratory results. The data collection tool was evaluated using 5% of the sample size. The lead investigator verified the accuracy and completeness of the data every day. Starting and ending controls were included in every batch of MTB culturing. The collected data were analyzed and interpreted accordingly after it was checked for completeness, accuracy, and clarity. The sterility of the culture media, sample processing reagents, and performance of the media were checked by incubating the whole media at 37 $^0$C for 48 hours, inoculating all reagents in a separate BHI and known susceptible *M. tuberculosis* (H37Rv), respectively. All laboratory results were recorded in a logbook and transformed into the data collection tool.

## Data analysis

The data was cleaned and entered in Epi data version 4.6 and exported to STATA version 17 software for analysis. The yield of contact investigation at each cascade was depicted using a processed map. Descriptive statistics were used to summarize data. Spear's man correlation analysis was done to determine the correlation between the index case ct value and the respective HHC. Binary logistic regression analysis was done at two levels. Groups were compared using a Chi-square test with Yates correction of continuity. Variables with P-values of 0.25 in bivariate logistic regression analysis were moved into multivariable logistic regression analysis. Finally, the adjusted odds ratio (aOR) with 95% confidence intervals and P-value < 0.05 was considered as statistically significant.

## Ethical consideration

The study was approved by Addis Ababa University, Aklilu Lema Institute of Pathobiology, with approval number ALIP IRERC/94/2015/23 and Ethiopian Public health Institute, EPHI-IRB-456-2022. The Addis Ababa Health Bureau also provided an official letter of support for the study. Following oral and written explanations of the study, voluntary index patients, HHCs, and guardians (when the index case or the HHC was children aged less than 15 years) signed written informed consent and assent. Individual records were closely protected in confidence, and data anonymization was ensured by aggregate analysis. Treatment of bacteriologically confirmed active TB patients (HHCs) and administration of TPT to children fifteen years of age or under who do not have active TB were conducted by the national TB guidelines.

## Result

A total of 335 bacteriologically confirmed PTB index patients were reached within the study period, and 303 of them were enrolled by fulfilling the study's inclusion criteria. The main reason index patients were ineligible was that they declared no HHC. Of the included index patients, 259 were DS-TB patients, while 44 were MDR-TB patients. A total of 967 HHCs were reported by the included index patients. The average number of HHCs per index case was 3.19(range 1–7). MDR-TB index cases had more HHCs per index case than DS-TB index cases (4.59 and 2.95, respectively) (Fig 1).

During household visits, a total of 902(93.07%) HHCs had received baseline symptom-based TB evaluation. The proportion of HHCs screened among the declared total HHCs was lower in contacts of MDR-TB index cases than in DS-TB index cases (79.70% and 96.86%, respectively). Presumptive tuberculosis was identified in 182 (20.17%) of the evaluated HHCs. All presumptive HHcs produced sputum samples for TB laboratory diagnosis. In the DS-TB and MDR-TB index case groups, 11 (1.48%) of 765 HHCs and 2 (1.24%) of 161 HHCs were diagnosed with active tuberculosis, respectively. All the 13 active tuberculosis cases found from HHCs were culture positive. This gave 1.44% TB prevalence among HHCs enrolled in the study (Fig 1).

The median age and BMI of TB symptom-screened HHCs were 28 years and 20.57, respectively. Among the screened HHCs, 513(56.83%) were female, 92(10.20%) were under 15 years children, and 167(18.51%) were under weight. The most prevalent symptom was a cough of any duration, 214 (23.73%). According to the national TB case management guideline, only 182 (20.17%) HHCs had presumptive TB. The most common symptom among TB symptom screening positive HHCs was cough with any duration 214 (23.73%) followed by fever with any duration 79 (8.76%). Among household contacts of index patients, 167(18.51%) were underweight, 206(22.89%) had BCG vaccination scars on their hands and 92 (10.20%) had comorbid disease. Only, 43 (4.77%) HHCs reported that they either currently smoke or had a history of smoking cigarettes (Table 1).

The proportion of total TB symptom screened HHCs among declared, presumptive TB, bacteriologically confirmed active pulmonary TB cases diagnosed varied by sex, age, and

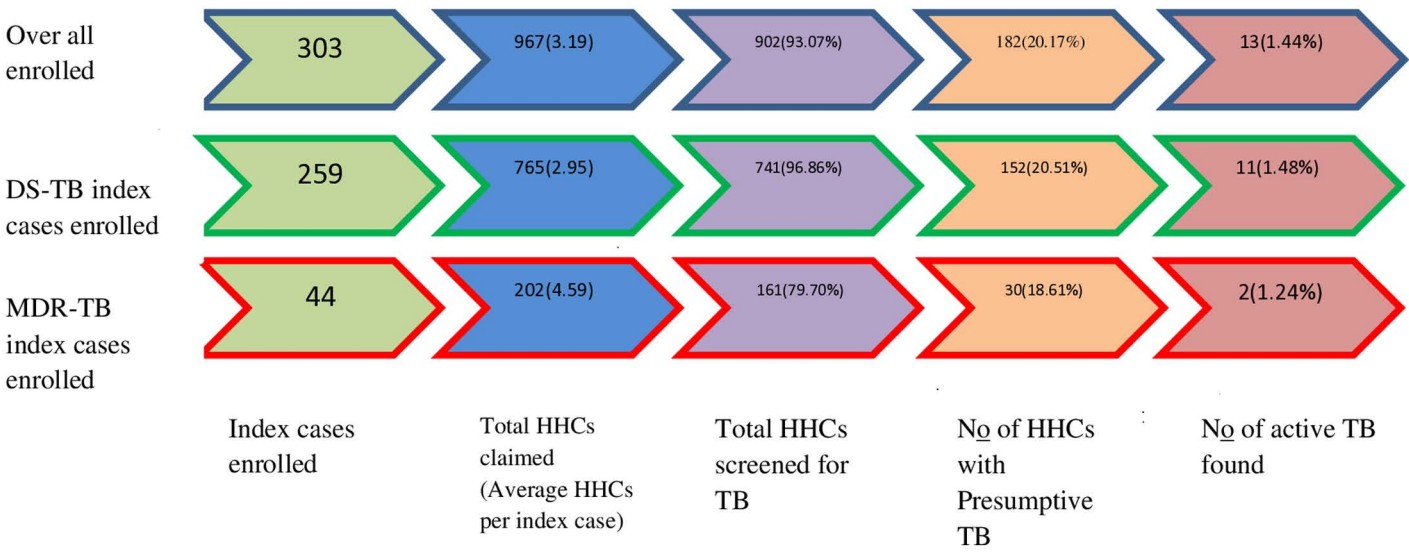

**Fig 1. Home –to-home TB contact investigation cascade in central Ethiopia from January 1 to December 3, 2023.**

**Table 1. Characterstics of bacteriologically confirmed pulmonary index cases and their TB screened HHCs in central Ethiopia from January 1 to December 30, 2023.**

| Characteristics | | Index case; N(%) = 303 | HHCs; N (%) = 902 |
|---|---|---|---|
| Sex | Male | 168(55.45) | 389(43.17) |
| | Female | 135 (44.55) | 513(56.83) |
| Age | Median | | 28(20–38) |
| | <15 | 7 (2.31) | 92(10.20) |
| | 16–24 | 70 (23.10) | 248(27.49) |
| | 25–34 | 114 (37.62) | 269(29.82) |
| | 35–44 | 60 (19.80) | 144 (15.96) |
| | 45–65 | 40(13.20) | 139 (15.41) |
| | >65 | 12 (3.96) | 10 (1.11) |
| BMI | Under weight(<18.5) | NA | 167(18.51) |
| | Normal weight(18.5–24.9) | NA | 709(78.60) |
| | Over weight(>25.0) | NA | 26 (2.88) |
| Educational status | Illiterate | 36(11.88) | 54 (5.99) |
| | Read and write | 15(4.95) | 26 (2.89) |
| | Primary | 70 (23.10) | 155 (17.20) |
| | Secondary | 97 (32.01) | 354 (39.18) |
| | Certificate and above | 85 (28.05) | 313 (34.74) |
| Route transmission know | Yes | NA | 591 (65.52) |
| | No | NA | 311 (34.48) |
| Relation with Index | Husband | NA | 80 (8.88) |
| | Wife | NA | 64 (7.10) |
| | Child | NA | 242 (26.75) |
| | Other | NA | 233 (25.86) |
| | Relative | NA | 283 (31.41) |
| BCG vaccinated(have scar) | Yes | NA | 206(22.89) |
| | No | NA | 696 (77.11) |
| cough > 2 weeks | Yes | NA | 182(20.17%) |
| | No | NA | 720(79.83%) |
| Cough any duration | Yes | NA | 214 (23.73) |
| | No | NA | 688 (76.27) |
| Fever any duration | Yes | NA | 79 (8.76) |
| | No | NA | 823 (91.24) |
| Weight loss | Yes | NA | 26 (2.88) |
| | No | NA | 876 (97.12) |
| Night sweeting | Yes | NA | 32 (3.55) |
| | No | NA | 870 (96.45) |
| Chest pain | Yes | NA | 9 (1.00) |
| | No | NA | 893 (99.00) |
| HIV status | Positive | 45 (14.85) | 21 (2.33) |
| | Negative | 255 (84.16) | 388 (43.02) |
| | Unknown | 3 (0.99) | 493(54.66) |
| Have underline medical condition | Yes | NA | 92 (10.20) |
| | No | NA | 810 (89.80) |
| Time with Index | All time | NA | 507(56.21) |
| | Night | NA | 256 (28.38) |
| | Day | NA | 129 (14.30) |
| | Other | NA | 10 (1.11) |

*(Continued)*

**Table 1.** (Continued)

| Characteristics | | Index case; N(%) = 303 | HHCs; N (%) = 902 |
|---|---|---|---|
| Drink alcohol daily | Yes | 44 (14.52) | 120 (13.30) |
| | No | 259 (85.48 | 782 (86.70) |
| History/Current smoker | Yes | 23 (7.59) | 43 (4.77) |
| | No | 280 (92.41) | 859 (95.23) |
| Share bed | Yes | NA | 208 (23.06) |
| | No | NA | 694 (76.94) |

NA=Not applicable/no data.

index drug susceptibility type. There were overall differences in the proportions of presumptive TB cases by age groups(<15 years old,15.21%;15–24 years,27.12%;25–34 years,19.33%;35–44 years,14.58%;45–65 years,23.02% and > 65 years,60.0%;p-value = 0.007). Even though it was not statistically significant (p-value = 0.45) presumptive TB cases differ by sex (27.12% in males and 23.91% in females). The proportions of bacteriologically confirmed active TB cases by age groups were (<15 years old,0.00%;15–24 years,2.82%;25–34 years,1.11%;35–44 years,0.69%;45–65 years,1.43% and > 65 years,0.00%;p-value = 0.34) (Fig 2).

The following risk factors were found to associate with risk of presumptive TB after adjustment for these potential confounders: age > 65 (aOR(95%CI), 15.15(2.44–94.11), illiterate educational status (aOR(95%CI), 2.41(1.08–5.93), relation with index other

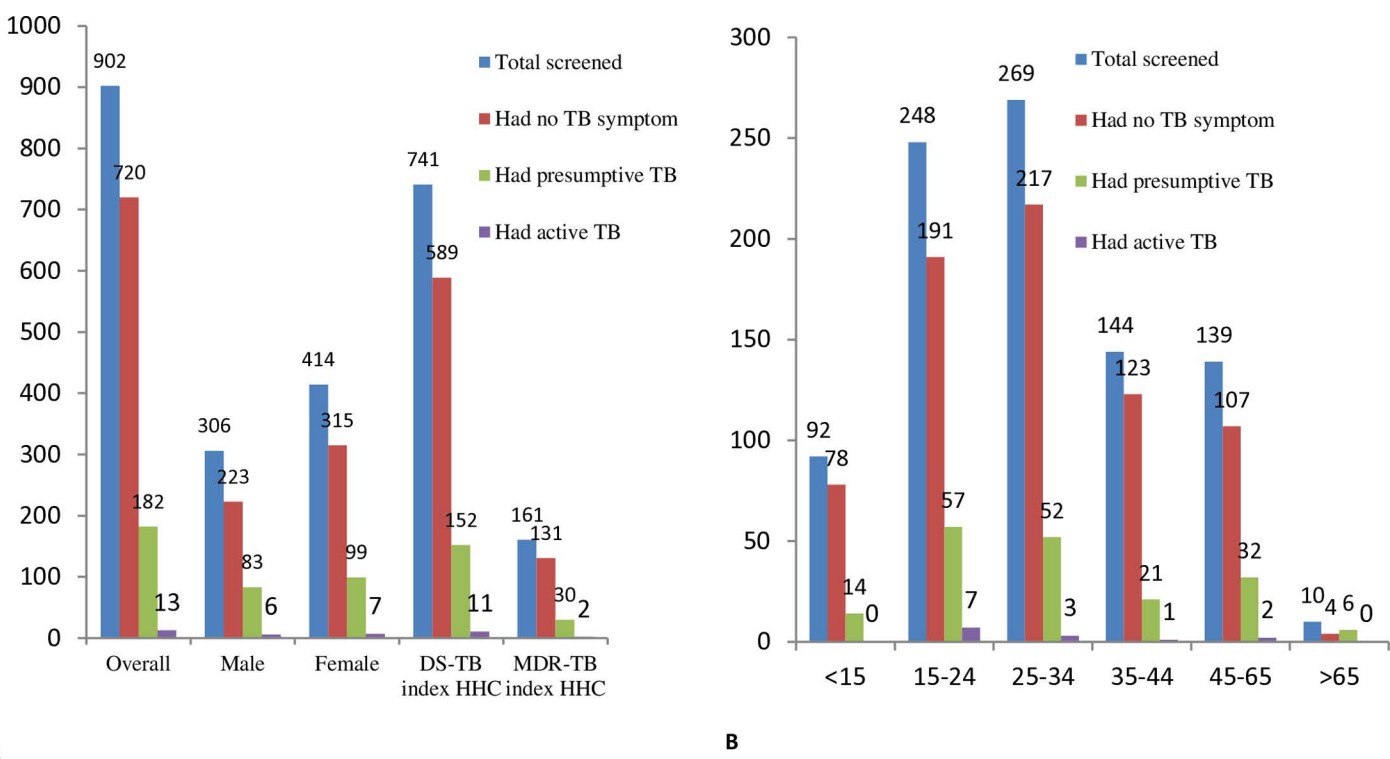

**Fig 2. Proportion of Asymptomatic, presumptive TB and active TB in HHCs: by A. sex, index Drug resistance type and overall B. age group in central Ethiopia from January 1 to December 3, 2023.**

than parents (aOR(95%CI), 0.50(0.28–0.91),having underline medical condition(aOR(95%CI),1.99(1.08–3.66), night time exposure to index case (aOR(95%CI),0.33(0.22–0.49) day time exposure to index case(aOR (95%CI), 0.44(0.22–0.86) (Table 2). Active TB

**Table 2. Comparative characteristics of tuberculosis symptom screening positive and screening negative household contacts at base line visit in central Ethiopia from January 1 to December 30, 2023.**

| Variable | | Symptomatic household contacts (n = 182(20.17%)) | Asymptomatic household contacts (n = 740(79.83%)) | AOR(95%CI) | P-value |
|---|---|---|---|---|---|
| Sex | Female | 99(19.30) | 414 (80.70) | 1 | |
| | Male | 83(21.34) | 306(78.66) | 1.13(0.74–1.71) | 0.55 |
| Age | <15 | 14 (15.22) | 78 (84.78) | 1 | |
| | 16–24 | 57 (22.98) | 191(77.02) | 2.29(0.96–5.45) | 0.06 |
| | 25–34 | 52 (19.33) | 217 (80.67) | 2.55(0.90–7.21) | 0.07 |
| | 35–44 | 21 (14.58) | 123 (85.42) | 1.53(0.49–4.72) | 0.45 |
| | 45–65 | 32 (23.02) | 107 (76.98) | 2.64(0.89–7.76) | 0.07 |
| | >65 | 6 (60.00) | 4 (40.00) | 15.15(2.44–94.11) | **0.01** |
| BMI | Under weight(<18.5) | 40 (23.95) | 127 (76.05) | 1 | |
| | Normal weight(18.5–24.9) | 139 (19.61) | 570 (80.39) | 0.70(0.42–1.17) | 0.18 |
| | Over weight(>25.0) | 3 (11.54) | 23 (88.46) | 0.47(0.11–1.97) | 0.30 |
| Educational status | Primary | 29(18.71) | 126 (81.29) | 1 | |
| | Illiterate | 16 (29.63) | 38 (70.37) | 2.41(1.08–5.93) | **0.05** |
| | Read and write | 8 (30.77) | 18 (69.23) | 2.31(0.76–6.95) | 0.13 |
| | Secondary | 66 (18.64) | 288 (81.36) | 0.82(0.44–1.50) | 0.52 |
| | Certificate and above | 63 (20.13) | 250 (79.87) | 0.90(0.45–1.80) | 0.78 |
| Route of transmission know | Yes | 127(21.49) | 464 (78.51) | 1.30(0.83–2.02) | 0.23 |
| | No | 55 (17.68) | 256 (82.32) | 1 | |
| Relation with Index | Child | 52 (21.49) | 190 (78.51) | 1 | |
| | Husband | 19 (23.75) | 61 (76.25) | 0.76(0.34–1.69) | 0.51 |
| | Wife | 21 (32.81) | 43 (67.19) | 1.82(0.83–3.97) | 0.13 |
| | Other | 27 (11.59) | 206 (88.41) | 0.50(0.28–0.91) | **0.02** |
| | Relative | 63 (22.26) | 220 (77.74) | 1.14(0.70–1.85) | 0.59 |
| BCG vaccinated | Yes | 47 (22.82) | 159 (77.18) | 1.06(0.60–1.86) | 0.83 |
| | No | 135 (19.42) | 560 (80.58) | 1 | |
| HIV status | Negative | 115(29.64) | 273(70.36) | 1 | |
| | Positive | 10 (47.62) | 11 (52.38) | 1.45(0.51–4.10) | |
| | Unknown | 57 (11.56) | 436 (88.44) | 0.33(0.22–0.49) | **0.01** |
| Underline medical condition | Yes | 31 (33.70) | 61 (66.30) | 1.99(1.08–3.66) | **0.02** |
| | No | 151(18.64) | 659 (81.36) | 1 | |
| Time with Index | All time | 120 (23.67) | 387 (76.33) | 1 | |
| | Night | 46(17.97) | 210 (82.03) | 0.64(0.41–0.99) | **0.04** |
| | Day | 13 (10.08) | 116 (89.92) | 0.44(0.22–0.86) | **0.01** |
| | Other | 3 (30.00) | 7 (70.00) | 1.25(0.27–5.67) | 0.77 |
| Drink alcohol daily | Yes | 25 (20.83) | 95 (79.17) | 0.83(0.43–1.59) | 0.58 |
| | No | 157 (20.08) | 625 (79.92) | 1 | |
| Current smoker | Yes | 13 (30.23) | 30 (69.77) | 2.20(0.89–5.39) | 0.08 |
| | No | 169 (19.67 | 690 (80.33) | 1 | |
| Share bed | Yes | 50 (24.04)) | 158 (75.96) | 1.05(0.65–1.70) | 0.82 |
| | No | 132 (19.02) | 562 (80.98) | 1 | |

positivity was significantly associated with night sweating, chest pain, any duration of fever, weight loss, night sweat and sharing a bed with the index patient, P-value < 0.05 (Table 3).

Among the DS-TB index case contacts, 152 (20.51%; 95% CI: 17.76–23.57) were diagnosed with presumptive TB, and from these 11 (7.24%; 95% CI: 3.85–12.9%) were found to be MTB detected Rifampin resistance not detected cases by Xpert Ultra assay and culture test. Whereas among thirty presumptive household contacts of MDR-TB patients, 2 (6.67%; 95% CI: 1.16–23.51) were MTB detected Rifampicin resistance was not detected. Rifampicin resistance active PTB was not detected in both study groups. The Xpert ultra assay results revealed an 84.62% (95% CI: 57.77–95.68) rifampicin drug resistance concordance between the index case and the corresponding household contact (Fig 3).

A Spearman correlation analysis was performed to determine the relationship between the Xpert MTB/RIF probe cycle thresh hold (ct) values of the index case and their respective HHCs. There was a positive correlation between the ct values of the index case and the corresponding HHC ct value at each probe. This analysis indicates a substantial correlation between the ct values of the index cases and the corresponding HHC, indicating that there is an association between the TB index ct values and corresponding HHc ct values that might be related to TB transmission. There is an increased risk of TB transmission to close contacts when the TB index case has a high bacillary load (Fig 4).

**Table 3. Risk factors for active tuberculosis in household contacts of bacteriologically confirmed pulmonary tuberculosis patients in central Ethiopia from January 1 to December 30, 2023.**

| variable | | Active TB diagnosed (%) | Without active TB (%) | Chi-Square test | P-value[*] |
|---|---|---|---|---|---|
| Total | | 13/902(1.44%) | 889/902(98.56) | | |
| HHCs sex | Male | 6 (1.54) | 383 (98.46) | 0.61 | 0.61 |
| | Female | 7 (1.36) | 506 (98.64) | | |
| Route of transmission know | Yes | 9 (1.52) | 582 (98.48) | 0.08 | 0.99 |
| | No | 4 (1.29) | 307 (98.71) | | |
| BCG vaccinated | Yes | 5 (2.43) | 201 (97.57) | 0.01 | 0.99 |
| | No | 8 (1.15) | 687 (98.85) | | |
| Any duration fever | Yes | 3 (3.80) | 76 (96.20) | 14.56 | <0.001 |
| | No | 10 (1.22) | 813 (98.78) | | |
| Weight loss | Yes | 3 (11.54) | 23 (88.46) | 36.64 | <0.001 |
| | No | 10 (1.14) | 866 (11.54) | | |
| Night sweat | Yes | 4 (12.50) | 28 (87.50) | 69.97 | <0.001 |
| | No | 9 (1.03) | 861 (98.97) | | |
| Chest pain | Yes | 2 (22.22) | 7 (77.78) | 27.63 | <0.001 |
| | No | 11 (1.23) | 882 (22.22) | | |
| Have underline medical condition | Yes | 2 (2.17) | 90 (97.83) | 0.38 | 0.87 |
| | No | 11 (1.36) | 799 (98.64) | | |
| Drink alcohol daily | Yes | 1 (0.83) | 119 (99.17) | 0.05 | 0.99 |
| | No | 12 (1.53) | 770 (98.47) | | |
| History/Current smoker | Yes | 1 (2.33) | 42 (97.67) | 3.27 | 0.24 |
| | No | 12 (1.40) | 847 (98.60) | | |
| Share bed | Yes | 7 (3.37) | 201 (96.63) | 3.96 | 0.05 |
| | No | 6 (0.86) | 688 (99.14) | | |

[*]Yates continuity correction P-value.

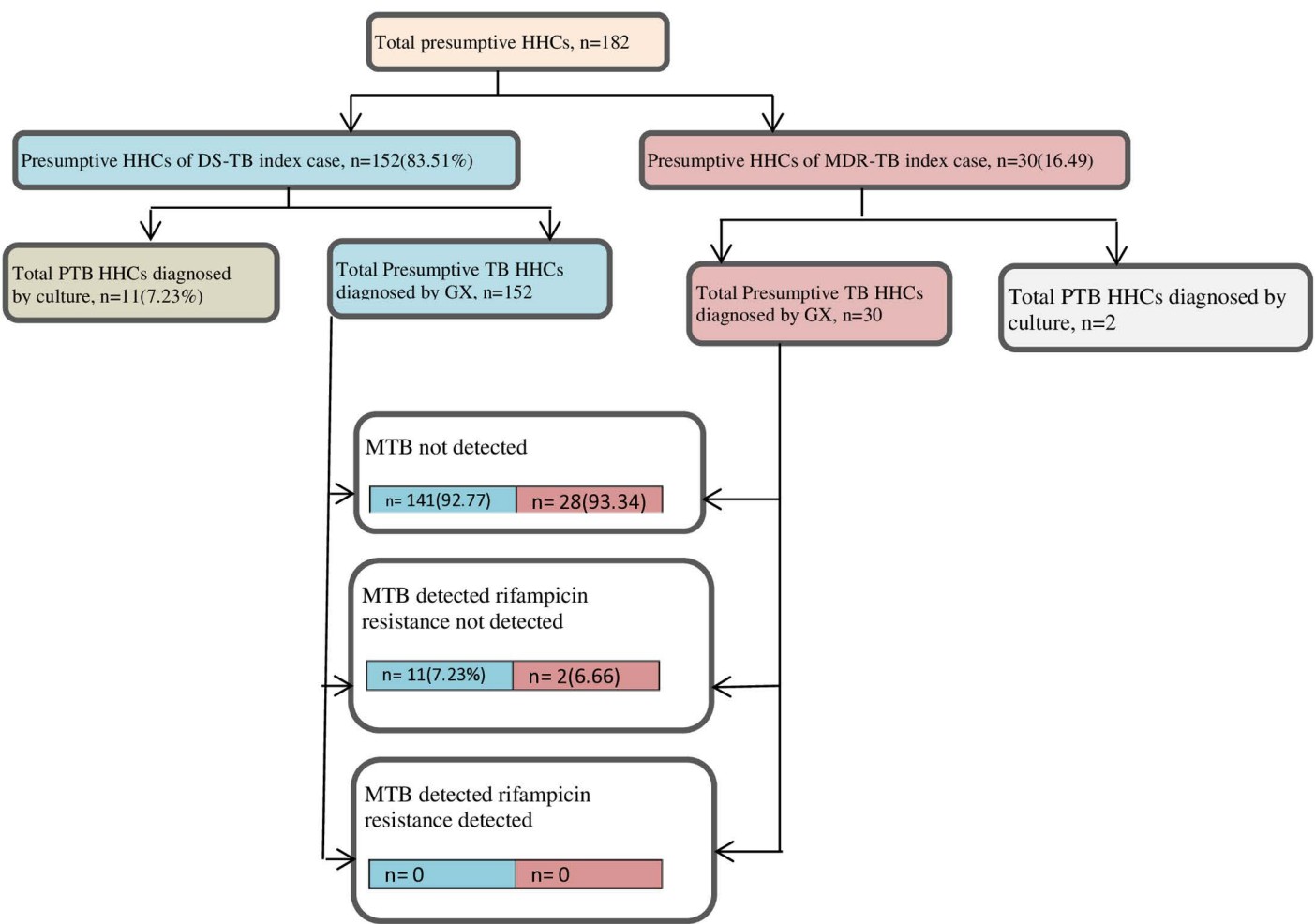

**Fig 3. Bacteriological confirmation of household contacts having active pulmonary TB from January 1 to December 3, 2023.** Purple color is for MDR-TB index case and Blue color is for DS-TB index case.

## Discussion

Household contacts of patients with bacteriologically confirmed pulmonary tuberculosis are significantly at higher risk of contracting active tuberculosis. To curve tuberculosis trend under any setting, evaluation of these household contacts under any setting should be a crucial component of the public health response. This study found that home-to-home TB contact investigation had a higher active TB yield than the routine TB contact investigation implemented in similar health facilities [21]. This indicates that scheduling convenient times and last-mile service delivery to contacts is very important to address the missed active TB cases in the community. To the best of our knowledge, this is the first door-to-door active case finding of all age group household contacts of patients with bacteriologically confirmed DS-TB or MDR-TB in central Ethiopia. Using home-to-home active TB case screening in central Ethiopia, we found, on average, 144 HHCs with active TB per 10000 household contacts screened. We noticed that booking a convenient time (at least two times, one is for those who missed the first visit) for all HHC members increases screening adherence and therefore yields active TB.

Our study reported a yield of 1.44% active TB amongst HHCs of bacteriologically confirmed pulmonary TB index cases that were screened using the Ministry of Health-approved

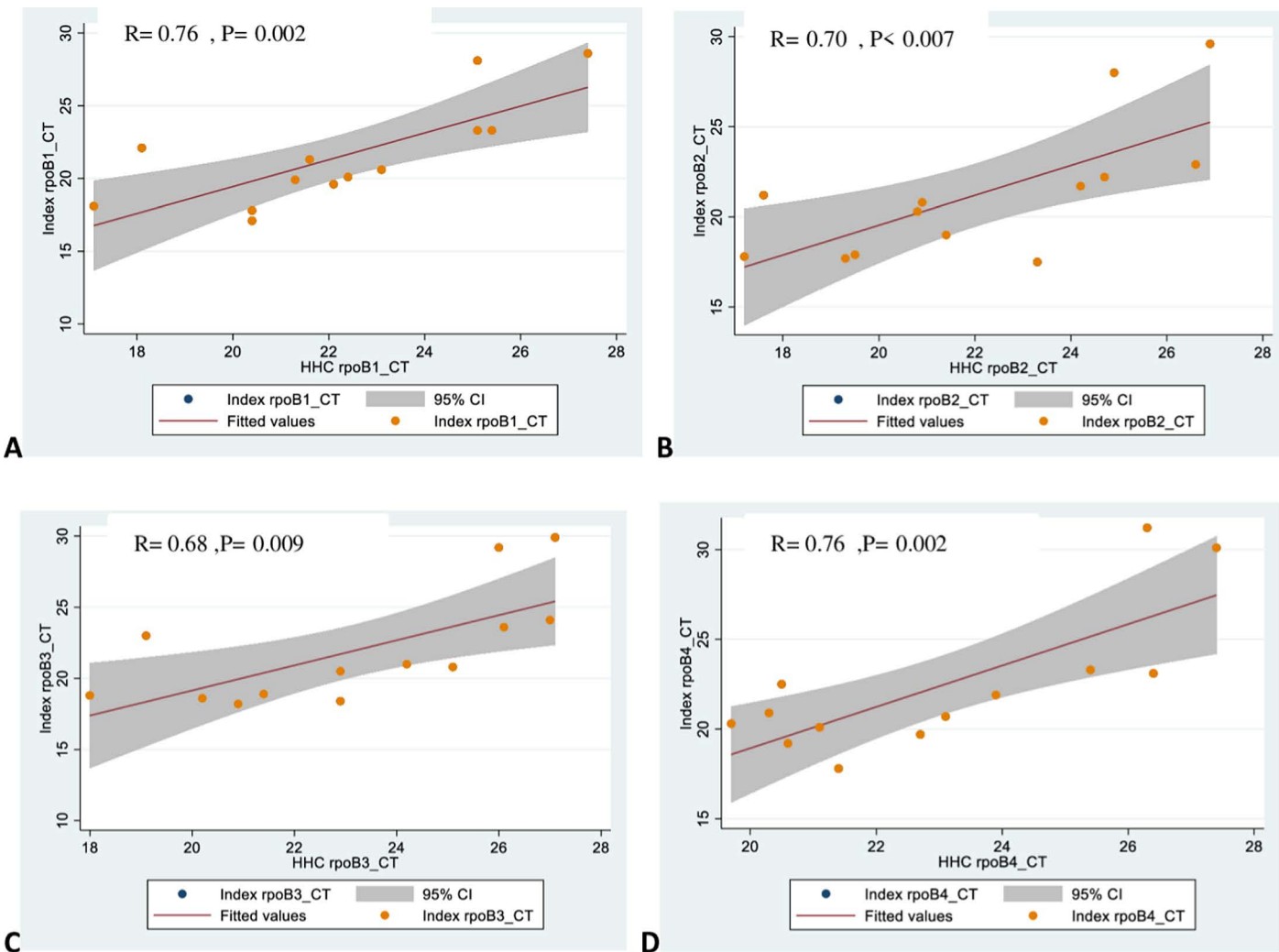

**Fig 4. Spearman correlation between Xpert CT parameters of index cases and HHCs.** (A). Xpert CT rpoB1; (B). Xpert CT rpoB2; (C). Xpert CT rpoB3 and (D). Xpert CT rpoB4.

symptom screening method only. Previous studies have reported different proportions of active tuberculosis from HHC investigations. Our result is similar to that documented in a study conducted in Nepal, Guniea, and China which reported 1.6% [22], 1.6% [23], and, 1.7% [24], respectively. It was higher than the reports from Vietnam (1.0%) [25], India (1.15%) [26], Iran (1.1%) [27], and in Ethiopia (1.1%) [18]. However, it was lower than the community TB screening finding from Uganda which reported 2.35% [28], and from Indonesia 2.4% [29]. This variation might be attributed to differences in the study populations, and study settings including TB burden, household ventilation, and sleeping arrangements, community living habits, health-seeking behavior of household contacts, infectiousness of index cases, vulnerability of contacts, and study methodology, with differences in sample size, screening algorithm, and diagnostic accuracy. The screening algorithm (only symptomatic HHC were privileged to give sputum samples without support by chest x-ray) used in our investigation might affect the yield of active tuberculosis.

Although the number of MDR-TB index patients in our study was fewer than the number of DS-TB index cases, we found no significant difference in active tuberculosis contact

investigation yield among HHCs based on the drug resistance profile of the index case. A similar study finding was reported from Ethiopia [30]. MDRTB patients are less likely to infect others than patients with drug-susceptible tuberculosis [31]. This makes our finding welcome but it lacked statistical power to distinguish between the number of secondary cases in MDR-TB index cases HHCs versus DS-TB index case contacts.

The yield of HHCs investigation is impacted by several factors, and it deteriorates when it comes to children's contact. In this study, there was a significant prevalence of presumptive TB in children (aged less than 15 years), but no active TB cases were detected. Our study findings were in contradiction with studies from Tajikistan [32] Myanmar [33] and Pakistan [34] which reported active TB cases from children's household contacts. In our study, the majority of the index cases were not parents; children were more proxies for their parents than other family members, potentially affecting the yield of active tuberculosis in children. Even though we were capable of collecting samples from all presumptive cases, the study's sample-collecting strategy did not enhance the production of suitable samples from children. It suggests the need for additional techniques to take appropriate samples from children. After tuberculosis was ruled out, preventive therapy was commenced for all children based on the eligibility criteria.

In our study, we found a high uptake for TB symptom screening (93.07%) and testing for active TB among HHCs. A similar finding was reported from South Africa(95% screened at baseline) [35]. Engaging full-time, salaried, and familiar with the community HEWs in TB active-case finding schemes might help us to achieve much-appreciated achievement in this target. To get high symptom screening uptake scheduling a convenient time when most of the HHCs are available is important MDR-TB index case's HHcs had lower TB symptom screening than DS-TB index case's HHCs. This might be the fact that relatively DOT facilities were closer to patients' residences than MDR-TB TICs. Around one–fifth of the HHCs were symptomatic for PTB, this result was lower than the finding from Myanmar (39%) [33] but it was higher than the reports from Ethiopia (13%) [18] Delivery of TB screening to the door of HHCs might increase the need of the contacts to be tested which forced them to falsely report the symptoms.

The risk factors (sharing a bed with an index patient and night sweetening) for active tuberculosis revealed in this study were comparable to those from the Chinese study [29,36]. The risk of contracting TB from an infectious patient increases as the proximity and duration of exposure to the index case increases [37]. Even though the exact cause of night sweating is still unknown, it is a vital sign that should not be disregarded while screening for tuberculosis [38].

The Xpert MTB/RIF Ultra assay results from this study indicate there was a high concordance of rifampicin drug resistance (84.62%) between the index case and the corresponding household contact. This finding was similar to the result of the meta-analysis pooled estimate [39] which had reported the pooled Pooled isoniazid/rifampicin concordance was 82.6% and report from Pakistan [10]. In a single household, it is possible for two individuals to have been exposed to different M. tuberculosis strains from the surrounding community. Studies have described that a significant number of contacts living in the same household possess different drug-resistance profiles in contrast to the index patients [10,39]. However, it must be supported by a whole genome sequencing test.

The significant strength of the study was its pragmatic nature, which employed the existing system established by the national tuberculosis program, healthcare professionals work in the health facility and HEWs work in the community. This indicates that home-to-home TB contact screening was effective and implementable in both rural and urban areas of the nation only by mentoring and motivating the health extension workers. To ensure early detection and

treatment of TB, healthcare providers must screen HHCs of patients with bacteriologically confirmed TB in timely manner.

One of the limitations of our study is the possibility of over-diagnosis which results from HHCs falsely reporting TB symptoms by considering as a great chance of being diagnosed at home. We did not use clinical screening like chest X-rays which might increase the sensitivity of screening. Additionally, we were unable to screen all the listed household contacts which may have resulted in the estimation of active TB yield. Financial and resource shortages limited us to testing latent TB infection in the HHCs. We suggest conducting additional research using whole genome sequencing to identify the actual index case and understand the transmission dynamics.

## Conclusion

Home-to-home TB contact screening have high active TB yield and implementable in both rural and urban areas of the nation only by mentoring and motivating the health extension workers. Proximity to bacteriologically confirmed TB patient for long time exposes household contacts for active TB. Incorporating door-to-door TB investigations into existing public health structures, such as utilizing health extension workers and women's community groups, could improve TB control strategies. Scheduling convenient times and last-mile service delivery to contacts is very important to address the missed active TB cases in the community. To combat this catastrophic infectious disease we have to strength our strategy and leave the reluctance.

## Supporting information

**S1 Fig. Tuberculosis screening algorithm for HHCs.**
(DOCX)

## Acknowledgments

We are deeply grateful to Addis Ababa University, Addis Ababa Health Bureau, and each site Health Office for providing the necessary ethical approval and support letters to conduct this study. Our heartfelt gratitude also goes to the Ethiopian Public Health Institute for providing guidance, the laboratory testing facility, and the ethical approval needed to conduct this study. Finally, we would like to strongly acknowledge the study participants, health workers, and health extension workers at each study sites.

## Author contributions

**Conceptualization:** Getachew Seid.

**Data curation:** Getachew Seid, Getu Diriba, Michael Hailu, Amanuel Wondimu, Mengistu Tadesse.

**Formal analysis:** Getachew Seid, Ayinalem Alemu, Balako Gumi.

**Investigation:** Getachew Seid, Getu Diriba, Amanuel Wondimu.

**Methodology:** Ayinalem Alemu, Getu Diriba, Mengistu Tadesse, Solomon H Mariam, Balako Gumi.

**Project administration:** Michael Hailu, Solomon H Mariam.

**Resources:** Gemechu Tadesse, Balako Gumi.

**Supervision:** Ayinalem Alemu, Mengistu Tadesse, Gemechu Tadesse, Solomon H Mariam.

**Writing – original draft:** Getachew Seid.

**Writing – review & editing:** Getachew Seid, Ayinalem Alemu, Michael Hailu, Amanuel Wondimu, Mengistu Tadesse, Gemechu Tadesse, Solomon H Mariam, Balako Gumi.

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
