## [Decision Letter · Decision Letter 0]

5 Nov 2024

PONE-D-24-35318Active Tuberculosis in Household contacts of Bacteriologically confirmed Pulmonary Tuberculosis patients: A Multicenter Study Finding the ‘Missed One' in Central Ethiopia.PLOS ONE

Dear Dr. Abegaz,

Thank you for submitting your manuscript, PONE-D-24-20230, entitled "Active Tuberculosis in Household contacts of Bacteriologically confirmed Pulmonary Tuberculosis patients: A Multicenter Study Finding the ‘Missed One' in Central Ethiopia" to PLoS One. Your work has been reviewed by experts in the field, who recognize the importance of your research and its potential value to readers. However, the reviewers raised some questions regarding your submission. After careful consideration, we feel that it has merit and therefore, we invite you to submit a revised version of the manuscript that addresses the points raised during the review process.

We look forward to receiving your revised manuscript.

Kind regards,

Ravi Prakash

Academic Editor

PLOS ONE

Journal Requirements:

3. Please include captions for your Supporting Information files at the end of your manuscript, and update any in-text citations to match accordingly. Please see our Supporting Information guidelines for more information: http://journals.plos.org/plosone/s/supporting-information .

Additional Editor Comments:

Reviewer 1

Minor Revision

This manuscript provides very good information and clarifies further gap in the low economic countries like Ethiopia. The manuscript can be considered for publication after required minor corrections given below. The manuscript is meeting the PONE requirements. Statistical analysis was executed properly. All the required data provided in tales and supplementals.

Reviewer 2

Major Revision

The manuscript addresses an important public health issue by examining active TB detection through door-to-door household contact investigations in Ethiopia. It highlights a significant gap in active TB case detection and provides insights into how home-based interventions might improve detection rates among high-risk groups. But the study design to result is not new, just add a piece of evidence to the established public health knowledge.

Specific Comments:

1.Introduction:

References such as those on TB incidence and the 'missing millions,' could benefit from more recent data to reflect any changes in global trends.

2. Methods:

1)The study mentions using the Xpert Ultra assay and TB culture for diagnosis. It would be helpful to briefly describe the sensitivity and specificity of these methods.

2)Consider adding more details on how "presumptive TB" was defined during the household evaluations.

3)Actually its Spearman's rank correlation coefficient or Spearman's ρ, not "Spear’s man correlation".

3. Results:

1)The section detailing the correlation between the index case’s cycle threshold (Ct) values and household contacts is intriguing. It would be beneficial to include a brief explanation of why this correlation is important in the context of TB transmission.

2)The comparison between DS-TB and MDR-TB households is valuable. However, the discussion of the similar yield in active TB detection between the two groups should address the potential for misclassification or under-diagnosis of MDR-TB in the HHCs.

3)Consider highlighting key findings in bullet points or using summary tables for improved clarity.

4)Figure 1 is useful in showing the contact investigation cascade. However, it could be enhanced with a visual distinction between DS-TB and MDR-TB groups to make differences more apparent.

4. Discussion:

1)The limitations are acknowledged, such as the lack of clinical screening (e.g., chest X-rays). However, the authors could further discuss how the use of such diagnostic tools might alter the study's conclusions.

2)The study's impact on future TB control strategies could be emphasized more. For example, what specific recommendations can be made for integrating door-to-door TB investigations into existing public health frameworks?

Reviewers' comments:

Reviewer's Responses to Questions

**Comments to the Author**

1. Is the manuscript technically sound, and do the data support the conclusions?

Reviewer #1: Yes

Reviewer #2: Partly

2. Has the statistical analysis been performed appropriately and rigorously? 

Reviewer #1: Yes

Reviewer #2: Yes

3. Have the authors made all data underlying the findings in their manuscript fully available?

Reviewer #1: Yes

Reviewer #2: Yes

4. Is the manuscript presented in an intelligible fashion and written in standard English?

Reviewer #1: Yes

Reviewer #2: Yes

5. Review Comments to the Author

Reviewer #1: This manuscript provides very good information and clarifies further gap in the low economic countries like Ethiopia. The manuscript can be considered for publication after required minor corrections given below. The manuscript is meeting the PONE requirements. Statistical analysis was executed properly. All the required data provided in tales and supplementals.

Reviewer #2: The manuscript addresses an important public health issue by examining active TB detection through door-to-door household contact investigations in Ethiopia. It highlights a significant gap in active TB case detection and provides insights into how home-based interventions might improve detection rates among high-risk groups. But the study design to result is not new, just add a piece of evidence to the established public health knowledge.

Specific Comments:

1.Introduction:

References such as those on TB incidence and the 'missing millions,' could benefit from more recent data to reflect any changes in global trends.

2. Methods:

1)The study mentions using the Xpert Ultra assay and TB culture for diagnosis. It would be helpful to briefly describe the sensitivity and specificity of these methods.

2)Consider adding more details on how "presumptive TB" was defined during the household evaluations.

3)Actually its Spearman's rank correlation coefficient or Spearman's ρ, not "Spear’s man correlation".

3. Results:

1)The section detailing the correlation between the index case’s cycle threshold (Ct) values and household contacts is intriguing. It would be beneficial to include a brief explanation of why this correlation is important in the context of TB transmission.

2)The comparison between DS-TB and MDR-TB households is valuable. However, the discussion of the similar yield in active TB detection between the two groups should address the potential for misclassification or under-diagnosis of MDR-TB in the HHCs.

3)Consider highlighting key findings in bullet points or using summary tables for improved clarity.

4)Figure 1 is useful in showing the contact investigation cascade. However, it could be enhanced with a visual distinction between DS-TB and MDR-TB groups to make differences more apparent.

4. Discussion:

1)The limitations are acknowledged, such as the lack of clinical screening (e.g., chest X-rays). However, the authors could further discuss how the use of such diagnostic tools might alter the study's conclusions.

2)The study's impact on future TB control strategies could be emphasized more. For example, what specific recommendations can be made for integrating door-to-door TB investigations into existing public health frameworks?

6. PLOS authors have the option to publish the peer review history of their article (what does this mean? ). If published, this will include your full peer review and any attached files.

**Do you want your identity to be public for this peer review?** For information about this choice, including consent withdrawal, please see our Privacy Policy .

Reviewer #1: **Yes: ** Ramgopal Sivanadham

Reviewer #2: No

---

## [Author Response · Author response to Decision Letter 1]

13 Nov 2024

Response to reviewers

Manuscript Number: PONE-D-24-35318

Title: Active Tuberculosis in Household contacts of Bacteriologically confirmed Pulmonary Tuberculosis patients: A Multicentre Study Finding the ‘Missed One' in Central Ethiopia.

Dear Editor and Reviewers

We would like to thank you for your constructive and in-depth review and comments. We believe that the given comments and suggestions improved the quality of the paper.

Specific Comments:

1.Introduction:

References such as those on TB incidence and the 'missing millions,' could benefit from more recent data to reflect any changes in global trends.

Response: Thank you for the comment, we put a reference in the revised manuscript. Yes the data was updated based on the recent WHO global TB report .Page 1 line 72-73

2.Methods:

1)The study mentions using the Xpert Ultra assay and TB culture for diagnosis. It would be helpful to briefly describe the sensitivity and specificity of these methods.

Response: Thank you for the suggestion. In the revised manuscript, we described the sensitivity and specificity of Xpert Ultra assay and the lower detection limit for TB culture was included in method section page 6 line 199-202.

“Xpert Ultra assay has a sensitivity of 90%(95%CI:84-94) and specificity 96%(95%CI:93-98). Tuberculosis culture is highly sensitive with lower detection limit of 10-100CFU/ml”

2) Consider adding more details on how "presumptive TB" was defined during the household evaluations.

Response: Thank you for the important comment. We revised it, based on the given comment (line 181-184).

3) Actually its Spearman's rank correlation coefficient or Spearman's ρ, not "Spear’s man correlation".

Response: Thank you for the comment, we revised it accordingly.

3. Results:

1)The section detailing the correlation between the index case’s cycle threshold (Ct) values and household contacts is intriguing. It would be beneficial to include a brief explanation of why this correlation is important in the context of TB transmission.

Response: Thank you for the important suggestion. It is corrected accordingly. There is an increased risk of Mtb transmission to close contacts when the TB index case has a high bacillary load. Page 12 line 310-311

2)The comparison between DS-TB and MDR-TB households is valuable. However, the discussion of the similar yield in active TB detection between the two groups should address the potential for misclassification or under-diagnosis of MDR-TB in the HHCs.

Response: Thank you for the comment. We tried to discuss the finding in line 341-347.

Although the number of MDR-TB index patients in our study was fewer than the number of DS-TB index cases, we found no significant difference in active tuberculosis contact investigation yield among HHCs based on the drug resistance profile of the index case. A similar study finding was reported from Ethiopia (30). MDRTB patients are less likely to infect others than patients with drug-susceptible tuberculosis (31). This makes our finding welcome but it lacked statistical power to distinguish between the number of secondary cases in MDR-TB index cases HHCs versus DS-TB index case contacts.

3) Consider highlighting key findings in bullet points or using summary tables for improved clarity.

Response: Thank you for the suggestion. We tried to put key findings of the study at the end of result section. Page 12 line 309-315

4) Figure 1 is useful in showing the contact investigation cascade. However, it could be enhanced with a visual distinction between DS-TB and MDR-TB groups to make differences more apparent.

Response: We revised the figure as per the given comment and now the visualization is improved.

4. Discussion:

1)The limitations are acknowledged, such as the lack of clinical screening (e.g., chest X-rays). However, the authors could further discuss how the use of such diagnostic tools might alter the study's conclusions.

Response: The chest x–ray might increase the sensitivity of TB screening as it increase the number of presumptive TB case. It was addressed in page 13 line 379.

2)The study's impact on future TB control strategies could be emphasized more. For example, what specific recommendations can be made for integrating door-to-door TB investigations into existing public health frameworks?

Response: Incorporating door-to-door TB investigations into existing public health structures, such as utilizing health extension workers and women's community groups, could improve TB control strategies.

Please see conclusion part of revised manuscript page 14

This manuscript provides very good information and clarifies further gap in the low economic countries like Ethiopia. The manuscript can be considered for publication after required minor corrections given below.

1. Abbreviate the short words like TB, HHC in its first appearance especially in Abstract.

Response: Thank you for the suggestion and we made a revision accordingly

2. In the abstract background section, please provide the clear intro and gaps in the area of research. The statement “The study was conducted to assess the yield of door-to-door TB household contact investigation among household contact of bacteriologically confirmed pulmonary TB cases in central Ethiopia” is better fit in to Methods section. A brief about the study gap is missing in the background section.

Response: Thank you for the important comment. We included the current gap in the abstract background section of the revised manuscript.

3. In Introduction section please correct the word contracted. “Worldwide in 2022, 10.6 million people contracted tuberculosis (TB)”. Should it be Contacted?

Response: Yes it was corrected with appropriate word

4. Please clarify on Household items you are targeting. What are the potential items in the Introduction session?

Response: We have targeted household items like number of living room, number of windows in the household.

5. Please either provide reference for the procedure used to diagnose the TB or explain the procedure. Such as for Smear microscopy, Gene Xpert MTB/RIF Ultra, and/or culture test.

Response: Yes the references were there in the method section page (reference number 19 and 20)

6. You have clearly given the inclusion criteria for the study subjects. Please also provide the exclusion criteria with proper justification.

Response: Thank you for the comment and it was described in method section page 5 line 161-162.

7. Provide a clear definition for the index TB case. As the study is majorly focussed on index TB.

Response: Thank you for the comment and it was included in method section page 5 line 154-156.

8. Statistical p values may require correction. Suggesting to use corrected p value. Such as Yates correction or Bonferroni correction. This will reduce the bias in the data.

Response: Thank you for the important comment we addressed the comments as follow Groups were compared using a Chi-square test with Yates correction of continuity. A two-tailed p value of less than 0.05 was considered as statistically significant. See Result section page 11 Table 3 and method section

9. Provide the message or benefit of study findings to the changes required in patient treatment, prescription, and recommendations to the clinicians in the discussion section and brief in conclusion.

Response: To ensure early detection and treatment of TB, healthcare providers must screen HHCs of patients with bacteriologically confirmed TB in timely manner. see page 14 line 392-393

10. None of the study area is 100% complete. Please provide the open research question, how much of the part this current study covered, and what is the gap in study. Give a clue to the researchers in similar field about the gaps and further research requirement.

Response: We suggest conducting additional research using whole genome sequencing to identify the actual index case and understand the transmission dynamics.

Please see page 14 line 382-383.

---

## [Decision Letter · Decision Letter 1]

1 Dec 2024

PONE-D-24-35318R1Active Tuberculosis in Household contacts of Bacteriologically confirmed Pulmonary Tuberculosis patients: A Multicenter Study Finding the ‘Missed One' in Central Ethiopia.PLOS ONE

Dear Dr. Abegaz,

Thank you for submitting your revised manuscript to PLOS ONE. After careful consideration, the reviewers still has some concerns in the manuscript. We feel that it has merit but does not fully meet PLOS ONE’s publication criteria as it currently stands. Therefore, we invite you to submit a revised version of the manuscript that addresses the points raised during the review process.

We look forward to receiving your revised manuscript.

Kind regards,

Ravi Prakash

Academic Editor

PLOS ONE

**Journal Requirements:**

**Additional Editor Comments:**

Reviewer 1:

No Comments

Reviewer 2:

Minor Revision

1. The definition of diagnostic is undoubtable following the guideline in line 141-145. "An active pulmonary TB case is a TB case confirmed by either smear microscopy, Gene Xpert MTB/RIF Ultra, or a TB culture test. Bacteriologically confirmed pulmonary tuberculosis: case referred to a pulmonary TB patient with biological specimen positive by Acid-Fast Bacilli (AFB) smear microscopy, Xpert MTB/RIF assay or TB culture, indifference with drug susceptibility profile."

Now what worrying is the diagnosis of bacteriologically confirmed pulmonary tuberculosis in HHCs, which means that the test may not covered all of three methods of AFB, Xpert, and culture simultaneously, it is a "OR" in your diagnostic criteria. The results may include false positive of dead bacteria patients, on the basis of the screening produce also not included the chest X-ray, which may bias the results.

This should be discussed and should be put forward in limitation, not only simply narrate in limitation line 389-390 "We did not use clinical screening like chest X-rays which might increase the sensitivity of screening. "

2.Line 292-294 "Whereas among thirty presumptive household contacts of MDR-TB patients, 2 (6.67%; 95% CI: 1.16- 23.51) were MTB detected Rifampicin resistance was not detected." this need to discuss in depth, is any other evidence or report that the MDR-TB HHCs not infected with MDT-TB strains, it should be addressed and clarify the primary and secondary drug resistance in this case.

3. line 300 "There was a positive correlation between the ct values of the index case and the corresponding HHC ct) value at each probe ", This bracket seems superfluous.

4. last time comment "Consider highlighting key findings in bullet points or using summary tables for improved clarity.

Author Response: Thank you for the suggestion. We tried to put key findings of the study at the end of result section. Page 12 line 309-315"

Lets make it clear that authors could embeded the key findings in the article, not only list and summary in the back of results, which is a bit redundant and may overlap with the discussion. Author should also be associated with the journal's editorial policy and format.

Reviewers' comments:

Reviewer's Responses to Questions

**Comments to the Author**

1. If the authors have adequately addressed your comments raised in a previous round of review and you feel that this manuscript is now acceptable for publication, you may indicate that here to bypass the “Comments to the Author” section, enter your conflict of interest statement in the “Confidential to Editor” section, and submit your "Accept" recommendation.

Reviewer #1: All comments have been addressed

Reviewer #2: All comments have been addressed

2. Is the manuscript technically sound, and do the data support the conclusions?

Reviewer #1: Yes

Reviewer #2: Yes

3. Has the statistical analysis been performed appropriately and rigorously? 

Reviewer #1: Yes

Reviewer #2: Yes

4. Have the authors made all data underlying the findings in their manuscript fully available?

Reviewer #1: Yes

Reviewer #2: Yes

5. Is the manuscript presented in an intelligible fashion and written in standard English?

Reviewer #1: Yes

Reviewer #2: Yes

6. Review Comments to the Author

**Reviewer #1:**  All my recommendations in minor revision have been addressed. The manuscript can be proceeded for next steps in publication.

**Reviewer #2:**  1. The definition of diagnostic is undoubtable following the guideline in line 141-145. "An active pulmonary TB case is a TB case confirmed by either smear microscopy, Gene Xpert MTB/RIF Ultra, or a TB culture test. Bacteriologically confirmed pulmonary tuberculosis: case referred to a pulmonary TB patient with biological specimen positive by Acid-Fast Bacilli (AFB) smear microscopy, Xpert MTB/RIF assay or TB culture, indifference with drug susceptibility profile."

Now what worrying is the diagnosis of bacteriologically confirmed pulmonary tuberculosis in HHCs, which means that the test may not covered all of three methods of AFB, Xpert, and culture simultaneously, it is a "OR" in your diagnostic criteria. The results may include false positive of dead bacteria patients, on the basis of the screening produce also not included the chest X-ray, which may bias the results.

This should be discussed and should be put forward in limitation, not only simply narrate in limitation line 389-390 "We did not use clinical screening like chest X-rays which might increase the sensitivity of screening. "

2.Line 292-294 "Whereas among thirty presumptive household contacts of MDR-TB patients, 2 (6.67%; 95% CI: 1.16- 23.51) were MTB detected Rifampicin resistance was not detected." this need to discuss in depth, is any other evidence or report that the MDR-TB HHCs not infected with MDT-TB strains, it should be addressed and clarify the primary and secondary drug resistance in this case.

3. line 300 "There was a positive correlation between the ct values of the index case and the corresponding HHC ct) value at each probe ", This bracket seems superfluous.

4. last time comment "Consider highlighting key findings in bullet points or using summary tables for improved clarity.

Author Response: Thank you for the suggestion. We tried to put key findings of the study at the end of result section. Page 12 line 309-315"

Lets make it clear that authors could embeded the key findings in the article, not only list and summary in the back of results, which is a bit redundant and may overlap with the discussion. Author should also be associated with the journal's editorial policy and format.

7. PLOS authors have the option to publish the peer review history of their article (what does this mean? ). If published, this will include your full peer review and any attached files.

**Do you want your identity to be public for this peer review?** For information about this choice, including consent withdrawal, please see our Privacy Policy .

Reviewer #1: **Yes: ** Ramgopal Sivanadham

Reviewer #2: No

---

## [Author Response · Author response to Decision Letter 2]

6 Dec 2024

Response to Reviewers

Reviewer #1: All my recommendations in minor revision have been addressed. The manuscript can be proceeded for next steps in publication.

Response: Dear reviewer thank you for your in-depth comments and review and we are lucky to address your comments.

Reviewer #2: 1. The definition of diagnostic is undoubtable following the guideline in line 141-145. "An active pulmonary TB case is a TB case confirmed by either smear microscopy, Gene Xpert MTB/RIF Ultra, or a TB culture test. Bacteriologically confirmed pulmonary tuberculosis: case referred to a pulmonary TB patient with biological specimen positive by Acid-Fast Bacilli (AFB) smear microscopy, Xpert MTB/RIF assay or TB culture, indifference with drug susceptibility profile."

Now what worrying is the diagnosis of bacteriologically confirmed pulmonary tuberculosis in HHCs, which means that the test may not covered all of three methods of AFB, Xpert, and culture simultaneously, it is a "OR" in your diagnostic criteria. The results may include false positive of dead bacteria patients, on the basis of the screening produce also not included the chest X-ray, which may bias the results.

This should be discussed and should be put forward in limitation, not only simply narrate in limitation line 389-390 "We did not use clinical screening like chest X-rays which might increase the sensitivity of screening. "

Response: Thank you for your comment. In this study from presumptive HHc sample we did (AFB) smear microscopy, Xpert ultra assay and TB culture test simultaneously. Page 6 line 184-185.It was corrected as follows in the revised manuscript. All the 13 active tuberculosis cases found from HHCs were culture positive. line 245-246

2.Line 292-294 "Whereas among thirty presumptive household contacts of MDR-TB patients, 2 (6.67%; 95% CI: 1.16- 23.51) were MTB detected Rifampicin resistance was not detected." this need to discuss in depth, is any other evidence or report that the MDR-TB HHCs not infected with MDT-TB strains, it should be addressed and clarify the primary and secondary drug resistance in this case.

Response: Thank you for the good comment. Close contacts of MDR-TB patients are more likely to contract drug-resistant TB (DR-TB). Meanwhile, contradicting evidence has been reported from many studies regarding the risk of TB in close contact of drug-susceptible and MDR-TB patients (reference numbers 8,9 and 10). line 91-92.

In a single household, two individuals can have been exposed to different M. tuberculosis strains from the surrounding community. Studies have described that a significant number of contacts living in the same household possess different drug-resistance profiles in contrast to the index patients (10, 39). line 380-385

For the culture-positive isolates, due to resource shortage, we are considering performing phenotypic DST in the future.

3. line 300 "There was a positive correlation between the ct values of the index case and the corresponding HHC ct) value at each probe ", This bracket seems superfluous.

Response: Thank you for the comment. We have edited the unnecessary bracket in the revised version.line 300

4. last time comment "Consider highlighting key findings in bullet points or using summary tables for improved clarity.

Author Response: Thank you for the suggestion. We tried to put key findings of the study at the end of result section. Page 12 line 309-315"

Lets make it clear that authors could embeded the key findings in the article, not only list and summary in the back of results, which is a bit redundant and may overlap with the discussion. Author should also be associated with the journal's editorial policy and format.

Response: Thank you for the nice comment. Of course the key findings were addressed in the first paragraph of the discussion section. line 318-327

We had read the plos one journal editorial policy and format but we were not able to get the specific section where the key finding(Author summary) of the study should have to put like other plos journals (e.g plos complex system,plos neglected tropical disease submission guideline which recommend author summary next to the abstract section ).Based on the journal editorial policy we are not able to add author summary.

---

## [Decision Letter · Decision Letter 2]

18 Dec 2024

Active Tuberculosis in Household contacts of Bacteriologically confirmed Pulmonary Tuberculosis patients: A Multicenter Study Finding the ‘Missed One' in Central Ethiopia.

PONE-D-24-35318R2

Dear Dr. Abegaz,

We’re pleased to inform you that your manuscript has been judged scientifically suitable for publication and will be formally accepted for publication once it meets all outstanding technical requirements.

Kind regards,

Ravi Prakash

Academic Editor

PLOS ONE

Additional Editor Comments (optional):

Reviewers' comments:

Reviewer's Responses to Questions

**Comments to the Author**

1. If the authors have adequately addressed your comments raised in a previous round of review and you feel that this manuscript is now acceptable for publication, you may indicate that here to bypass the “Comments to the Author” section, enter your conflict of interest statement in the “Confidential to Editor” section, and submit your "Accept" recommendation.

Reviewer #2: All comments have been addressed

2. Is the manuscript technically sound, and do the data support the conclusions?

Reviewer #2: Yes

3. Has the statistical analysis been performed appropriately and rigorously? 

Reviewer #2: Yes

4. Have the authors made all data underlying the findings in their manuscript fully available?

Reviewer #2: (No Response)

5. Is the manuscript presented in an intelligible fashion and written in standard English?

Reviewer #2: Yes

6. Review Comments to the Author

Reviewer #2: All minor revision suggestions have been implemented, and the manuscript is now ready to move forward with the publication process.

7. PLOS authors have the option to publish the peer review history of their article (what does this mean? ). If published, this will include your full peer review and any attached files.

**Do you want your identity to be public for this peer review?** For information about this choice, including consent withdrawal, please see our Privacy Policy .

Reviewer #2: No

---

## [Editor Report · Acceptance letter]

PONE-D-24-35318R2

PLOS ONE

Dear Dr. Seid,

I'm pleased to inform you that your manuscript has been deemed suitable for publication in PLOS ONE. Congratulations! Your manuscript is now being handed over to our production team.

Kind regards,

on behalf of

Dr. Ravi Prakash

Academic Editor

PLOS ONE